# Prevalence of anemia among reproductive women in different social group in India: Cross-sectional study using nationally representative data

Nowaj Sharif[1], Bhaswati Das[1]*, Asraful Alam[2]

1 Centre for the Study of Regional Development, School of Social Sciences, Jawaharlal Nehru University (JNU), New Delhi, India, 2 Department of Geography, Serampore Girls' College, University of Calcutta, Kolkata, West Bengal, India

☯ These authors contributed equally to this work.

* bhaswati2004@gmail.com

## Abstract

### Background

The common cause of anemia in the general population is iron deficiency. Anemia is adversely affecting women of reproductive age and child health which in turn results in increased morbidity and maternal death, and also hamper social-economic growth. Reproductive women are more prone to anemia due to inadequate dietary intake and iron loss during menstruation and pregnancy.

### Objectives

This study examined the level and trend of anemia prevalence among the socially disadvantaged group (SC&ST, OBC) of women as compared to the other women (general) and identified the main responsible factors behind this.

### Data and methods

The data for this analysis has been taken from three rounds of National Family Health Survey (NFHS) conducted during 2005–2006 (NFHS 3), 2015–16 (NFHS 4) and 2019–21 (NFHS 5). Univariate and bivariate analyses were used to assess the level of anemia among reproductive age group women of different social groups. The regression model represents the relative risk of different confounding factors on the occurrence of anemia. GIS technique used for graphical representation of anemia prevalence rate among different social groups of women in different states of India.

### Result

In India more than 15 states belong to the high prevalence (>%55) of anemia among socially backward groups in 2019–21. The anemia prevalence was high (>55%) in all social groups (SC & ST, OBC, general) observed in 7 states in NFHS-3, 4 in NFHS-4 and 11 states in

**Data Availability Statement:** All relevant data are available at: http://rchiips.org/nfhs/ http://rchiips.org/nfhs/nfhs4.shtml http://rchiips.org/nfhs/NFHS-5Reports/NFHS-5_INDIA_REPORT.pdf.

**Funding:** This research did not receive any specific grant from funding agencies in the public, commercial, or not-for-profit sectors.

**Competing interests:** The authors have declared that no competing interests exist.

NFHS-5. The overall result reveals that the SC&ST women were more prone to any anemia than OBC and general women and the prevalence rate slightly increased from 2005–06 to 2019–21. Among all variables, economic status dominantly controls the anemia level in all social groups. Anemia prevalence of the poor and poorest group of general women were much worse than the women of richer and richest groups of SC&ST, OBC. The odds of women having anemia were lower among higher educated and urban women as compared to the non educated and rural women, irrespective of social group. The prevalence of anemia decreases with increased age of women and increases with the number of child bearing. All differences were statistically significant.

## Conclusions

The problem of iron deficiency remains a major issue in India, where the majority of the states (eastern, north-eastern and central) suffer from high anemia prevalence rate and it increases over time. It is observed that multiple socio-demographic factors ranging from poor economic and educational status, rural residence to higher childbearing of women are responsible for predicting anemia levels among the social groups of women in India. To eradicate this problem India should improve women's overall nutrition status and their income. Meanwhile, GOI should be more focused on the existing policies related to anemia and on their actual implementation on grassroots level.

## Introduction

The 21st century is considered as the most advanced era in terms of economic opportunities and healthcare facilities for the human beings. But the problem is that these opportunities are not equally accessible to everyone [1]. India remains in the poorest rank as a developing country by nearly any measure in the world, with a population of more than 1.21 billion [2]. However, in the last some decades there have been considerable improvements observed in most health indicators, including a decline in infant or child and maternal mortality rates and a drop in the fertility rate to a nearly below-replacement level [3]. In contrast, India remains in a poor situation concerning nutritional status. In the 2021 Global Hunger Index, India scored 27.5 with a rank of 101 (fall from 94 in 2020) out of 116 countries [4]. Another report on Global Food Policy surveyed (2022) by International Food Policy Research Institute observed an alarming scenario, where they predict approximately 73.9 million Indians will suffer from hunger by 2030 [5]. The overall situation of nutrition status in India is poor, whereas the situation of women's health conditions was much more adverse because of the existence of gender discrimination from birth and uneven distribution of health services [6–8]. Nutrition deficiencies such as protein, vitamin C, and iron push women towards anemia. On the other hand, studies have shown that lower caste women are more anemic than the other women because they are very low groomed to make decisions regarding their life, health, education, food allocation, and consumption. As a result, the high rate of the burden of anemia among women in India catastrophically reflects their poor health and socio-economic condition both within society and the household [9, 10].

Anemia is adversely affecting women of reproductive age and child health which in turn results in increased morbidity and maternal death, and also hamper social-economic growth [11–13]. Anemia is defined by the World Health Organization as a reduction in the proportion

of red blood cells or decline in the concentration of hemoglobin level or insufficient oxygen caring capacity to fulfill the physiological demand [14]. Anemia has different precipitating factors like genetic causes such as hemoglobinopathies; infections, such as malaria. The nutritional consideration includes iron deficiency on one hand and deficiencies of vitamins like A and $B_{12}$ and minerals such as copper on the other hand [15]. The most common cause of anemia in the general population is iron deficiency. An estimate by the World Health Organization (WHO) that around to over half a billion women or 29.9% of reproductive women aged 15–49 years were suffering from anemia in 2019 and most of them suffer due to iron deficiency [16, 17]. Reproductive and adolescent women are more prone to anemia due to insufficient dietary intake and iron loss during menstruation and pregnancy [18].

The burden of anemia is one of the major public health issues in the world. But the magnitude of problems is not equal in every country or for every group. In developing countries problem is enormous, specifically among women and young children whereas the developed countries are also affected but in lesser magnitude, approximately 6% as compared to 27% of adolescent girls in developing countries [19]. Among the developing countries prevalence of anemia in South Asia is the highest in the world and India has a prevalence of high iron deficiency anemia. It was estimated that about 50 to 70 percent of reproductive women are anemic in India [20, 21]

The consequence of anemia depends on the severity, the social group, and living conditions and it impairs mental and psychomotor development, reduces individual work performance, increases maternal and child morbidity and mortality. However, after reviewing literature it has been documented that different potential factors among women including rural residency [22, 23] younger age [24], lower education of women [25], lower women's empowerment [26], poorer economic condition [27, 28], lower nutrition status [29], higher childbearing [10] increases the chances to be anemic. Additionally, lower consumption of alcohol and use of contraceptive was shown to have an important protective effect against anemia [30, 31].

The existing research on women's anemia has mainly considered factors like pregnant and non-pregnant women, maternal age, wealth and education status, place of residence, etc [32, 33]. In developing countries like India, social group is an important traditional measure of social stratification, which was based on the respondents' self-identification as belonging to scheduled caste (SC), scheduled tribe (ST), other backward class (OBC), and general (which are not fall in the category of SC, ST and OBC). In India, more than sixty percent people belong to the most socially disadvantaged group (collectively SC, ST, and OBC) [34]. These are the people who are exposed to poor living conditions and consumed poor diets with limited access to health care. Their substance upon low-quality food and limited availability of iron supplements ultimately can lead to anemia [35]. Previous studies speculated that caste influences education, household wealth, and gender bias, which directly impact women's health and lead to morbidity and increase the mortality rate [36].

This study aims to determine the prevalence of anemia levels and factors associated with it among SC, ST, OBC, and other women in India. In particular, through a comparative assessment of levels and factors associated with anemia among SC, ST, OBC, and other women, we have investigated two objectives: first, to highlight the level and trend of anemia prevalence among the socially disadvantaged group of women as compared to the other women in India and second, to identify the factors affecting anemia among SC & ST, OBC women vis-à-vis other women to prioritise the policy for these categories of women.

## Data and method

The data for analysis has been taken from three rounds of National Family Health Survey (NFHS) conducted during 2005–2006 (NFHS 3), 2015–16 (NFHS 4) and 2019–21 (NFHS

5). NFHS-3 surveyed 124385 women; NFHS-4 surveyed 699,686 and, NFHS-5 surveyed 724115women.

## Study population

This study considered reproductive age group women of 15–49 years across the states in India. Here we have excluded the women with missing information on anemia level, and other covariates like place of residence, education level, household wealth, working status, etc. from all round. The state of Nagaland (anemia data not collected in NFHS-III due to local opposition) and all union territories (except Delhi) were excluded from the entire analysis as either it was not represented in all the surveys or had a very thin sample. After removing these samples 112478 from NFHS-3, 688896 from NFHS-4, and 714421 from NFHS-5 women of reproductive age group have been considered for final analysis.

Social group is the main focus of this study for which sample was collected based on the self-reported social group by the head of the household. Types of the social groups were scheduled caste (SC), scheduled tribe (ST), other backward class (OBC), and other castes (general). In this study, SC and ST are merged together to form one single category due to low sample size of individual categories.

## Dependent variables

The dependent variable used in this study is whether there is anemia or not. The NFHS identified three levels of anemia namel, mild with Hb level 10.0–11.9 g/dl, moderate with Hb level 7.0–9.9 g/dl, and severe with Hb level less than 7.0 g/dl. Similarly, any anemia was defined as the Hb level <12.0 g/dl.

## Independent variable

In this work, predictor variable is considered based on an extensive review of existing literature pertaining to the risk of occurrence of anemia in developing countries including India [10, 22, 31, 37–39]. The predictor variables include demographic, economic and behavioural factors like age, marital status, children ever born, contraceptive use, place of residence, work status, wealth quintile, BMI, intake of pulses, and and source of drinking water. The variables like pregnant or not and drinking alcohol are not used in this analysis due to thin sample size (Table 1).

## Statistical analysis

Univariate and, bivariate analyses were used to assess the level of anemia among reproductive age group women of different social groups (SC & ST, OBC, and General caste) by their background characteristics with applying appropriate sampling weight. In bivariate analysis, chi-square test was used to check the statistical significance of the differences in anemia prevalence across the demographic, social and behavioural characteristics. Binary logistic regression is used and we have provided an adjusted ratio with 95% confidence intervals. The regression model represents the relative risk of different confounding factors on the occurrence of anemia among reproductive age group women of different social categories. All the statistical analyses in this study were performed using STATA 16 (Stata Corp LP, College Station, Texas, USA).

## Graphical representation

State-level maps were drawn for all three rounds by using the GIS technique to show the spatial variation of anemia among different social groups of women aged 15–49 years. To show

**Table 1. Distribution of women in reproductive age group from different social groups by their background characteristics in India, 2005/06 to 2019/21.**

| Background characteristics | NFHS 3 | | | NFHS 4 | | | NFHS 5 | | |
|---|---|---|---|---|---|---|---|---|---|
| | SC & ST (n = 31490) | OBC (n = 46182) | General (n = 36593) | SC & ST (n = 206005) | OBC (n = 303830) | General (n = 162258) | SC & ST (n = 225012) | OBC (n = 310772) | General (n = 153027) |
| **Anemia** | | | | | | | | | |
| Severe | 3.65 | 3.15 | 2.84 | 2.13 | 2.09 | 1.53 | 5.12 | 4.57 | 4.37 |
| Moderate | 29.62 | 26.63 | 25.27 | 24.87 | 23.36 | 21.43 | 52.51 | 49.13 | 49.24 |
| Mild | 66.73 | 70.23 | 71.89 | 73.00 | 74.55 | 77.04 | 42.37 | 46.30 | 46.39 |
| **Place of residence** | | | | | | | | | |
| Urban | 23.36 | 29.94 | 41.58 | 24.95 | 34.52 | 0.46 | 24.47 | 32.22 | 43.82 |
| Rural | 76.64 | 70.06 | 58.42 | 75.05 | 65.48 | 0.54 | 75.52 | 67.77 | 56.17 |
| **Educational Status** | | | | | | | | | |
| No education | 54.23 | 43.56 | 25.68 | 35.73 | 28.40 | 16.27 | 29.12 | 22.64 | 13.62 |
| Primary | 14.52 | 15.08 | 14.69 | 13.81 | 12.24 | 11.16 | 13.39 | 11.24 | 9.95 |
| Secondary and Higher | 31.25 | 41.36 | 59.62 | 50.45 | 59.37 | 72.57 | 57.49 | 66.12 | 76.43 |
| **Age Group** | | | | | | | | | |
| 15–24 | 39.51 | 38.46 | 36.13 | 36.28 | 35.26 | 32.81 | 34.90 | 33.79 | 30.69 |
| 25–34 | 30.68 | 30.38 | 31.14 | 30.51 | 29.96 | 30.44 | 30.24 | 29.55 | 30.61 |
| 35+ | 29.80 | 31.16 | 32.74 | 33.20 | 34.78 | 36.75 | 34.86 | 36.65 | 38.70 |
| **Wealth** | | | | | | | | | |
| Poorest | 31.15 | 15.67 | 8.17 | 28.57 | 15.92 | 8.10 | 29.25 | 14.96 | 8.70 |
| Poor | 24.14 | 20.79 | 12.69 | 24.54 | 18.93 | 14.01 | 24.20 | 19.66 | 13.91 |
| Middle | 19.42 | 23.72 | 17.28 | 20.69 | 21.66 | 18.10 | 20.15 | 22.30 | 18.18 |
| Richer | 15.15 | 22.60 | 24.54 | 15.74 | 23.33 | 23.69 | 15.69 | 23.43 | 23.67 |
| Richest | 10.13 | 17.22 | 37.33 | 10.45 | 20.15 | 36.09 | 10.71 | 19.65 | 35.55 |
| **Current Work status** | | | | | | | | | |
| Not working | 54.43 | 60.82 | 73.85 | 69.90 | 76.75 | 80.73 | 69.62 | 75.18 | 79.65 |
| Working | 45.57 | 39.18 | 26.15 | 30.10 | 23.25 | 19.27 | 30.38 | 24.82 | 20.35 |
| **Marital Status** | | | | | | | | | |
| Never Married | 18.79 | 19.31 | 22.16 | 22.53 | 22.60 | 23.16 | 23.95 | 23.69 | 24.07 |
| Currently Married | 75.42 | 76.39 | 73.64 | 72.74 | 73.45 | 72.84 | 71.13 | 72.30 | 72.17 |
| Formally Married | 5.79 | 4.29 | 4.20 | 4.72 | 3.96 | 4.00 | 4.92 | 4.01 | 3.76 |
| **Contraceptive use** | | | | | | | | | |
| Not using | 57.60 | 56.56 | 52.79 | 59.32 | 60.53 | 56.45 | 50.39 | 49.90 | 49.03 |
| CP & IUD | 2.59 | 2.49 | 5.50 | 3.79 | 2.78 | 5.80 | 4.84 | 3.90 | 5.98 |
| Female Sterilisation | 30.75 | 32.47 | 27.85 | 29.45 | 28.92 | 25.53 | 30.58 | 31.00 | 25.22 |
| Other | 9.07 | 8.48 | 13.86 | 7.44 | 7.77 | 12.22 | 14.19 | 15.20 | 19.77 |
| **Pregnant** | | | | | | | | | |
| Nonpregnant | 94.24 | 94.64 | 95.66 | 95.30 | 95.39 | 96.09 | 96.06 | 96.15 | 96.79 |
| Pregnant | 5.76 | 5.36 | 4.34 | 4.70 | 4.61 | 3.91 | 3.94 | 3.85 | 3.21 |
| **Total Children ever born** | | | | | | | | | |
| No Children | 27.11 | 27.71 | 29.82 | 30.41 | 30.32 | 30.71 | 31.32 | 30.66 | 30.74 |
| One to two children | 27.63 | 30.62 | 36.38 | 35.47 | 38.87 | 44.02 | 37.14 | 41.01 | 46.59 |
| More than Two Children | 45.25 | 41.67 | 33.80 | 34.12 | 30.81 | 25.27 | 31.54 | 28.32 | 22.66 |
| **BMI** | | | | | | | | | |
| Underweight | 41.64 | 34.87 | 29.10 | 26.74 | 22.44 | 17.64 | 21.44 | 18.44 | 14.57 |

(*Continued*)

**Table 1.** (Continued)

| Background characteristics | NFHS 3 | | | NFHS 4 | | | NFHS 5 | | |
|---|---|---|---|---|---|---|---|---|---|
| | SC & ST (n = 31490) | OBC (n = 46182) | General (n = 36593) | SC & ST (n = 206005) | OBC (n = 303830) | General (n = 162258) | SC & ST (n = 225012) | OBC (n = 310772) | General (n = 153027) |
| Normal | 51.18 | 53.56 | 52.85 | 58.37 | 56.98 | 55.45 | 59.74 | 57.03 | 54.77 |
| Overweight | 5.81 | 9.00 | 13.47 | 11.64 | 15.45 | 19.54 | 14.24 | 17.90 | 21.77 |
| Obese | 1.37 | 2.56 | 4.58 | 3.24 | 5.12 | 7.37 | 4.57 | 6.62 | 8.90 |
| **Mass media Exposure** | | | | | | | | | |
| No exposure | 30.96 | 24.40 | 15.00 | 24.00 | 19.36 | 11.69 | 27.77 | 21.51 | 15.22 |
| Partial exposure | 56.07 | 56.12 | 58.81 | 68.63 | 71.27 | 76.07 | 66.10 | 70.49 | 74.49 |
| Full exposure | 12.97 | 19.47 | 26.19 | 7.37 | 9.37 | 12.24 | 6.14 | 8.00 | 10.28 |
| **Frequecy of eating pulses** | | | | | | | | | |
| Less than daily/ not at al | 54.61 | 48.29 | 39.26 | 58.44 | 55.36 | 51.05 | 53.40 | 49.28 | 48.48 |
| Daily | 45.38 | 51.70 | 60.73 | 41.55 | 44.63 | 48.94 | 46.59 | 50.71 | 51.51 |
| **Source of Drinking water** | | | | | | | | | |
| Improved Source | 83.59 | 87.41 | 91.87 | 90.12 | 93.12 | 94.34 | 92.9 | 94.98 | 95.67 |
| Non-improved Souerce | 16.40 | 12.58 | 8.13 | 9.88 | 6.87 | 5.65 | 7.1 | 5.01 | 4.3 |
| **Alcohol consumption** | | | | | | | | | |
| No | 94.30 | 98.78 | 99.31 | 97.39 | 99.27 | 99.49 | 98.48 | 99.58 | 99.68 |
| Yes | 5.70 | 1.22 | 0.69 | 2.61 | 0.73 | 0.51 | 1.52 | 0.42 | 0.32 |

geographical distribution, 'any anemia prevalence' is considered as that makes the distribution of sample across different social groups adequate. Any anemia is categorised into three similar categories for all social groups which include percentage of population with low level of anemia [<50], middle level of anemia [50–55], and high level of anemia [>55].

## Results

### Spatial distribution showing the prevalence of anemia among women of reproductive age by social group in India

From 2005–06 to 2019–21, the prevalence of anemia among different social groups of women in all states of India was demonstrated in Figs 1–3. In NFHS-3 (Fig 1) Out of 28 states, 16 states had a higher prevalence (more than 55%) of anemia among SC & ST women, 14 states had anemia prevalence rate more than 55% among OBC women, and 10 states had among general women. Similarly, for SC & ST, OBC and General categories 4, 6, and 4 states respectively had a moderate prevalence of anemia. The lowest percentage of anemia prevalence (< 50%) is majorly among the general category (14 states) followed by OBC (8 states) and SC & ST (8 states). Fig 2 represents the anemia prevalence during 2015–16 (NFHS-4) in 28 states of India. Out of 28 states, only 6 states had a higher prevalence (more than 55%) of anemia among general and OBC women in India. Whereas, among SC & ST women prevalence of anemia is more than 55% in 12 states. The prevalence of anemia was low (50%) in only 11 states among SC & ST women, 14 states among OBC, and 15 states among general women in NFHS-4. Fig 3 represents the anemia prevalence of 2019–21 (NFHS-5), which significantly increases the percentage of anemic women. Out of 28 states, 16 states had a higher prevalence of anemia (>55%) among SC & ST and OBC women, and 13 states among general women. Meanwhile, for SC & ST, OBC and general categories 8, 9 and 11 states respectively belong to the lower anemia (<50%) group. The anemia prevalence was high (>55%) in all social groups (SC & ST,

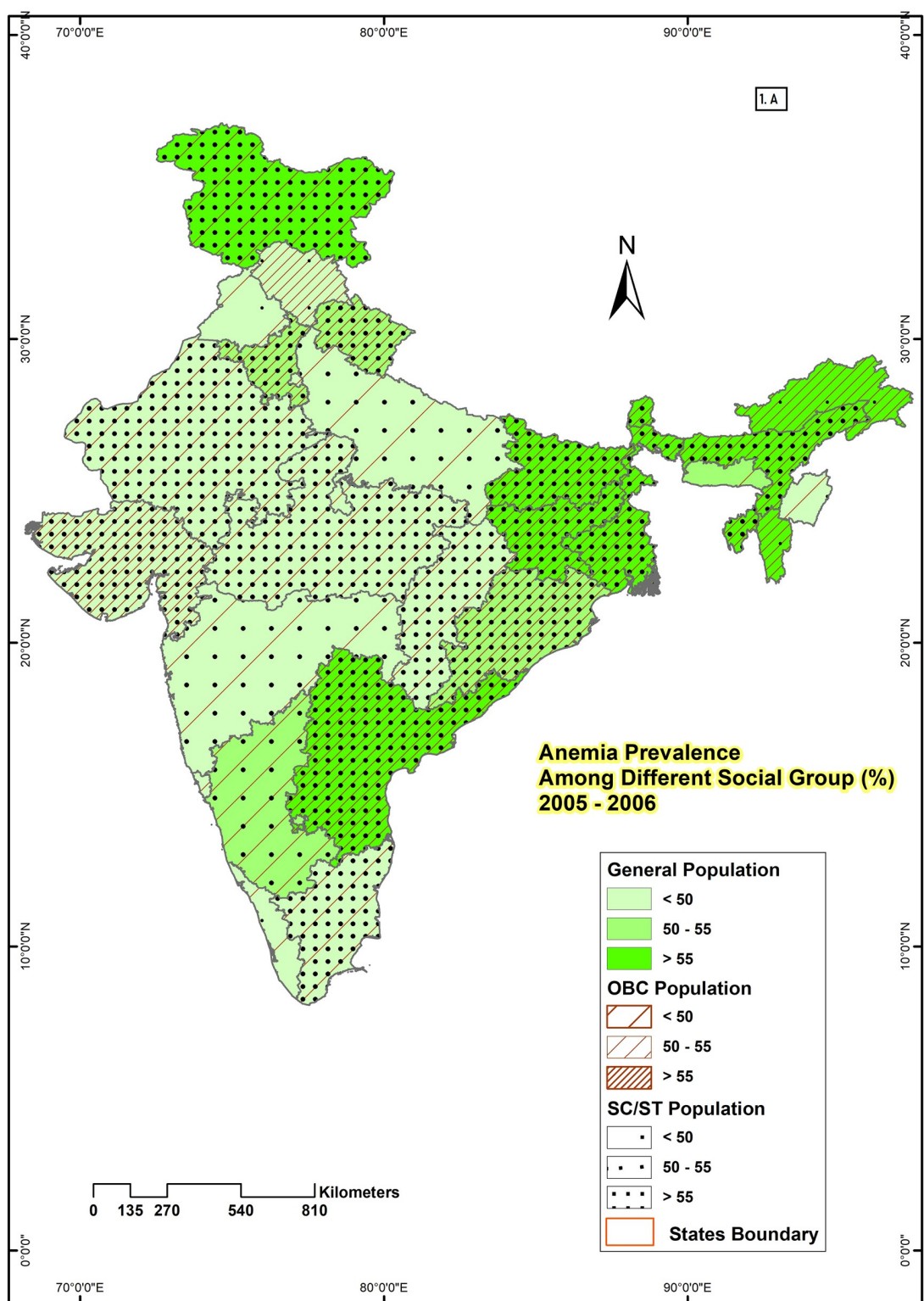

**Fig 1. State-wise variation in the prevalence of anemia among different social group of reproductive women in India, 2005–2006.**

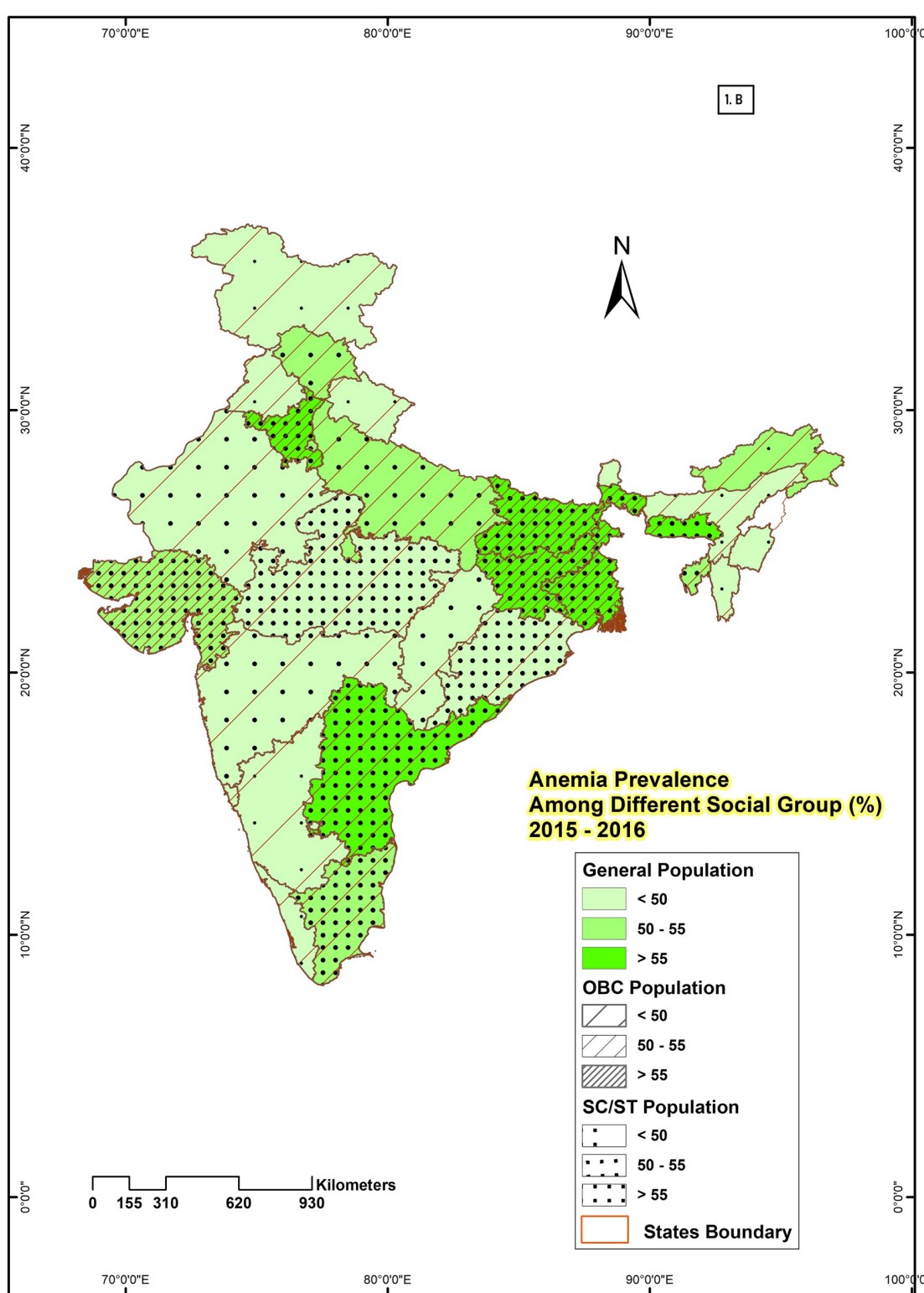

**Fig 2. State-wise variation in the prevalence of anemia among different social group of reproductive women in India, 2015–2016.**

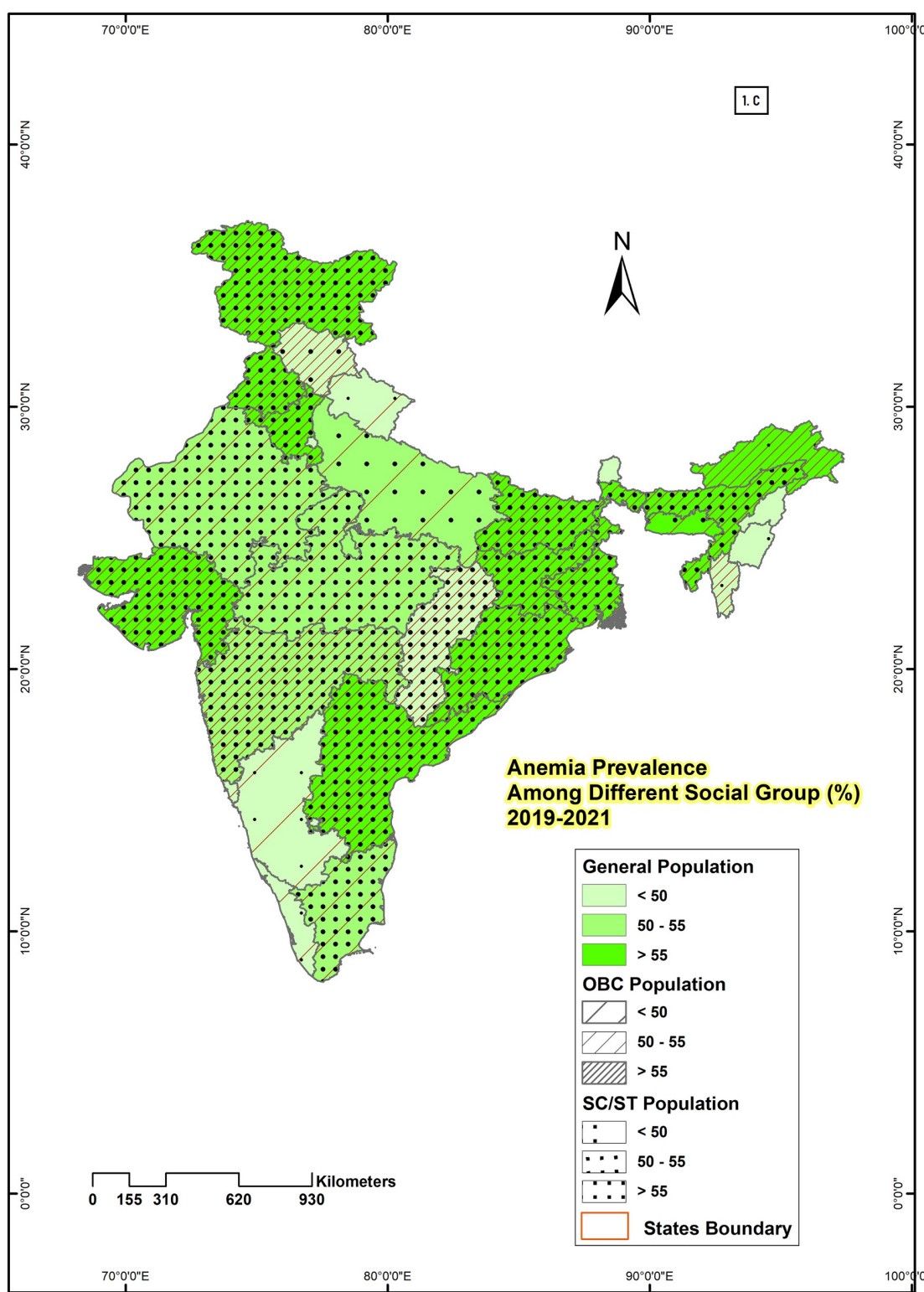

**Fig 3. State-wise variation in the prevalence of anemia among different social group of reproductive women in India, 2019–2021.**

OBC, general) observed in 7 states (Andhra Pradesh, Assam, Bihar, Jharkhand, Sikkim, Tripura & West Bengal) in NFHS-3, and 4 states (Bihar, Hariyana, Jharkhand, and West Bengal) in NFHS-4 and 11 states (Andhra Pradesh, Assam, Bihar, Gujrat, Haryana, Jammu and Kashmir, Jharkhand, Odisha, Punjab, Tripura, West Bengal) in NFHS-5. Jharkhand has the highest (81.22%) and Kerala (32.30%) had the lowest percentage of anemia prevalence among SC & ST women. Similarly among OBC and general women, the highest prevalence rate was observed in Assam and the lowest was in Punjab and Kerala in NFHS-3.

Mean while, the prevalence of anemia in NFHS-4 was highest in Jharkhand for SC & ST (72.05%) and OBC (61.90%) women, and for general women, the highest anemic state was Meghalaya (70.32%). Among the SC & ST and, OBC Manipur (22.99% & 28%), and among general women, Kerala (30%) had the lowest percentage of anemia prevalence. Similarly, West Bengal had the highest percentage of anemia prevalence for SC & ST (76.14%), and general (68.65%) women and Kerala experienced the lowest rate in NFHS-5.

## Baseline characteristics

Table 1 represents the baseline characteristics of the study population by their social group namely, SC & ST, OBC and others. Between three rounds, there is an increase in proportion of women in severe and moderate anemia categories across the social groups and there is commensurating decline in mild anemia with some exception during NFHS-4. Distribution of the study group population remained more or less consistent while considering the residential distribution, distribution by wealth quintiles and distribution by use of contraception. Rest all other have shown some inter-group variation in the distribution.

## Prevalence of anemia and its predictors

The bivariate analysis (Tables 2–4) illustrates the prevalence of anemia among reproductive age-group women of various social groups by different predictor variables. The overall result in all the rounds (NFHS 3, NFHS 4, and NFHS 5) reveals that women belonging to the SC & ST group has a higher prevalence of mild, moderate, and severe anemia (any anemia) as compared to the women of OBC and others. Meanwhile, the surveys reveal that the prevalence of anemia slightly increased or remained the same across all predictor variables irrespective of the social group from 2005/06 to 2019/21.

The difference in the prevalence of anemia persists among educational subgroups across SC & ST, OBC, and general. The prevalence rate reduced from 63.81% among uneducated SC & ST to 57.53% in secondary and higher-educated women, on the other hand in the general category a decline of 10 percentage points was observed between women with no education and women with higher and secondary education in NFHS-3. The surveys reveal that the prevalence of anemia declines in all social groups as their educational status improves. Expectedly, across the social categories and rounds the prevalence of anemia declined from the poorest to the richest wealth quintile. The poor women from the general caste are in the same boat where anemia prevalence is much higher than in other economic categories. Anemia prevalence in the poor and poorest group of general were much higher than it was among the richer and richest group of SC & ST and OBC (Tables 2–4). It has been observed that regular intake of pulses and improved sources of drinking water reduces the anemia prevalence across all the social groups. The chi-square value is highly significant across the surveys. However, anemia by age group and by the work status did not remain significantly consistent across the survey.

**Table 2. Prevalence of anemia among reproductive age-group women of different social groups by background characteristics in India, 2005–06.**

| Background characteristics | NFHS 3 | | | | | | | | | | | |
|---|---|---|---|---|---|---|---|---|---|---|---|---|
| | SC& ST | | | | OBC | | | | Others | | | |
| | Severe | Moderate | Mild | Any | Severe | Moderate | Mild | Any | Severe | Moderate | Mild | Any |
| *Age Group* | | | | | | | | | | | | |
| 15–24 | 2.33 | 19.46 | 40.96 | 62.75 | 1.68 | 15.04 | 38.16 | 54.88 | 1.31 | 13.41 | 37.30 | 52.02 |
| 25–34 | 2.05 | 17.07 | 41.56 | 60.68 | 1.58 | 14.47 | 37.52 | 53.57 | 1.35 | 13.12 | 36.67 | 51.13 |
| 35+ | 2.27 | 17.69 | 40.45 | 60.41 | 1.89 | 13.79 | 38.91 | 54.60 | 1.75 | 12.45 | 36.98 | 51.18 |
| χ2 | 30.01** | | | 17.12*** | 10.80* | | | 2.85 | 10.4* | | | 2.58 |
| *Place of Residence* | | | | | | | | | | | | |
| Urban | 2.45 | 17.63 | 36.69 | 56.77 | 1.54 | 13.26 | 35.94 | 50.74 | 1.10 | 12.02 | 35.37 | 48.49 |
| Rural | 2.16 | 18.37 | 42.30 | 62.83 | 1.79 | 15.00 | 39.17 | 55.96 | 1.73 | 13.70 | 38.16 | 53.59 |
| χ2 | 150.07*** | | | 146.75*** | 88.75*** | | | 80.78*** | 72.2*** | | | 35.3*** |
| *Educational Status* | | | | | | | | | | | | |
| No education | 2.30 | 19.33 | 42.18 | 63.81 | 1.96 | 16.08 | 40.12 | 58.15 | 2.12 | 16.12 | 39.79 | 58.03 |
| Primary | 2.14 | 17.73 | 40.97 | 60.84 | 2.13 | 15.46 | 38.62 | 56.21 | 1.85 | 14.37 | 37.72 | 53.95 |
| Secondary & Higher | 2.15 | 16.44 | 38.93 | 57.53 | 1.31 | 12.44 | 36.03 | 49.77 | 1.09 | 11.33 | 35.61 | 48.02 |
| χ2 | 325.31*** | | | 293.38*** | 225.81*** | | | 186.74*** | 325.86*** | | | 217.5*** |
| *Wealth Quintile* | | | | | | | | | | | | |
| Poorest | 2.30 | 20.70 | 45.40 | 68.40 | 1.82 | 16.50 | 41.14 | 59.46 | 1.90 | 16.98 | 43.11 | 61.98 |
| Poorer | 2.17 | 18.23 | 42.74 | 63.14 | 2.04 | 16.17 | 39.81 | 58.02 | 2.32 | 15.95 | 42.43 | 60.70 |
| middle | 2.46 | 16.87 | 38.56 | 57.88 | 2.23 | 15.46 | 38.17 | 55.86 | 1.89 | 14.73 | 38.17 | 54.79 |
| Richer | 2.24 | 17.16 | 36.87 | 56.27 | 1.46 | 12.99 | 37.42 | 51.88 | 1.38 | 12.85 | 36.53 | 50.76 |
| Richest | 1.68 | 14.50 | 34.09 | 50.27 | 0.84 | 11.18 | 34.65 | 46.67 | 0.94 | 10.43 | 33.58 | 44.95 |
| χ2 | 777.98*** | | | 743.49*** | 349.4*** | | | 284.85*** | 513.67*** | | | 424.4*** |
| *Current Work Status* | | | | | | | | | | | | |
| Not working | 2.18 | 18.04 | 41.11 | 61.33 | 1.54 | 14.10 | 37.81 | 53.45 | 1.36 | 12.64 | 37.19 | 51.19 |
| Working | 2.29 | 18.38 | 40.85 | 61.52 | 1.98 | 15.07 | 38.82 | 55.87 | 1.76 | 14.04 | 36.46 | 52.26 |
| χ2 | 4.32 | | | 2.60 | 3.97 | | | 3.62* | 2.66 | | | 0.05 |
| *Marital Status* | | | | | | | | | | | | |
| Not Married | 2.33 | 16.59 | 39.64 | 58.56 | 1.58 | 12.21 | 36.92 | 50.71 | 1.29 | 11.18 | 36.16 | 48.63 |
| Currently in union | 2.13 | 18.50 | 41.28 | 61.90 | 1.67 | 14.97 | 38.50 | 55.13 | 1.47 | 13.44 | 37.25 | 52.15 |
| Formaly Married | 3.26 | 19.40 | 41.66 | 64.32 | 3.14 | 15.87 | 38.78 | 57.79 | 2.25 | 15.03 | 37.12 | 54.39 |
| χ2 | 74.42*** | | | 51.47*** | 72.25*** | | | 38.88*** | 83.3*** | | | 60.33*** |
| *Total Children Ever Born* | | | | | | | | | | | | |
| No Children | 2.55 | 17.53 | 39.71 | 59.78 | 1.65 | 13.22 | 36.73 | 51.61 | 1.36 | 11.96 | 35.31 | 48.63 |
| 1 to 2 children | 2.37 | 19.52 | 41.26 | 63.15 | 1.87 | 14.93 | 37.61 | 54.41 | 1.33 | 12.60 | 36.64 | 50.56 |
| > 2 Children | 2.37 | 19.52 | 41.26 | 63.15 | 1.64 | 14.98 | 39.61 | 56.24 | 1.70 | 14.36 | 38.88 | 54.94 |
| χ2 | 39.63*** | | | 30.79*** | 52.86*** | | | 48.22*** | 133.43*** | | | 111.9*** |
| *BMI* | | | | | | | | | | | | |
| Underweight | 2.74 | 19.46 | 43.28 | 65.48 | 2.16 | 16.31 | 39.56 | 58.03 | 1.94 | 15.40 | 40.49 | 57.83 |
| Normal | 2.03 | 18.23 | 39.91 | 60.17 | 1.64 | 14.33 | 38.09 | 54.06 | 1.54 | 13.22 | 36.33 | 51.09 |
| Overweight | 0.61 | 10.33 | 34.82 | 45.76 | 0.68 | 9.77 | 35.18 | 45.63 | 0.51 | 8.47 | 32.79 | 41.77 |
| Obese | 1.04 | 11.71 | 38.28 | 51.03 | 0.83 | 9.18 | 32.64 | 42.65 | 0.42 | 8.60 | 34.95 | 43.97 |
| χ2 | 486.21*** | | | 431.76*** | 330.83*** | | | 261.54*** | 414.99*** | | | 336.8*** |
| *Contraceptive use* | | | | | | | | | | | | |
| Not using | 2.59 | 20.01 | 41.25 | 63.85 | 1.87 | 15.71 | 38.18 | 55.76 | 1.54 | 13.98 | 36.62 | 52.14 |
| Pill & IUD | 0.78 | 10.82 | 38.78 | 50.38 | 0.72 | 11.35 | 33.59 | 45.66 | 0.48 | 9.21 | 35.03 | 44.72 |
| Female Sterilization | 1.78 | 15.87 | 40.29 | 57.93 | 1.57 | 13.28 | 38.22 | 53.06 | 1.62 | 12.25 | 37.49 | 51.36 |

*(Continued)*

**Table 2.** (Continued)

| Background characteristics | NFHS 3 | | | | | | | | | | | |
| --- | --- | --- | --- | --- | --- | --- | --- | --- | --- | --- | --- | --- |
| | SC& ST | | | | OBC | | | | Others | | | |
| | Severe | Moderate | Mild | Any | Severe | Moderate | Mild | Any | Severe | Moderate | Mild | Any |
| Other | 1.86 | 16.65 | 42.39 | 60.90 | 1.48 | 11.78 | 39.66 | 52.92 | 1.26 | 12.30 | 38.24 | 51.80 |
| χ2 | 157.99*** | | | 99.38*** | 77.22*** | | | 29.17*** | 72.46*** | | | 17.86*** |
| *Mass Media Exposure* | | | | | | | | | | | | |
| No | 2.32 | 19.91 | 43.83 | 66.06 | 1.90 | 16.78 | 40.65 | 59.32 | 1.94 | 17.60 | 40.13 | 59.67 |
| Partial | 2.26 | 17.98 | 40.18 | 60.42 | 1.69 | 14.53 | 37.94 | 54.16 | 1.53 | 13.02 | 37.01 | 51.56 |
| everyday | 1.85 | 15.03 | 37.75 | 54.63 | 1.56 | 11.46 | 35.88 | 48.90 | 1.04 | 10.34 | 35.19 | 46.56 |
| χ2 | 381.3*** | | | 351.98*** | 194.4*** | | | 157.3*** | 325.75*** | | | 240.4*** |
| *Frequency of Eating Pulses* | | | | | | | | | | | | |
| Less than daily/not at al | 2.35 | 18.61 | 41.08 | 62.04 | 1.84 | 14.27 | 37.61 | 53.72 | 1.50 | 13.82 | 37.16 | 52.49 |
| Daily | 2.09 | 17.69 | 40.88 | 60.67 | 1.60 | 14.67 | 38.75 | 55.02 | 1.44 | 12.47 | 36.89 | 50.81 |
| χ2 | 8.00** | | | 6.83*** | 6.29*** | | | 4.97*** | 0.56 | | | 0.12 |
| Source of Drinking water | | | | | | | | | | | | |
| Improved Source | 2.21 | 17.67 | 40.24 | 60.12 | 1.7 | 14.45 | 37.95 | 54.10 | 1.46 | 13.10 | 36.79 | 51.35 |
| Non-improved Souerce | 2.60 | 20.34 | 44.23 | 67.17 | 1.61 | 14.23 | 39.12 | 54.96 | 1.44 | 12.69 | 38.25 | 52.38 |
| χ2 | 70.96*** | | | 69.97*** | 2.89 | | | 0.78 | 4.90 | | | 3.75** |

*$p < .10$.

**$p < .05$.

***$p < .01$.

## Adjusted odds ratio of anemia prevalence and its determinants among different social groups

Tables 5 & 6 represent the result of logistic regression analysis of anemia prevalence among SC & ST, OBC, and general women while controlling for background characteristics. The adjusted odds of anemia prevalence was found to be less among higher aged SC & ST (OR = 0.87 NFHS-3; OR = 0.86 NFHS-4; OR = 0.75 NFHS-5, p <0.01), OBC (OR = 0.9 NFHS-3; OR = 0.79 NFHS-4; OR = 0.87 NFHS-5 p <0.01) and general women (OR = 0.84 NFHS-3; OR = 0.88 NFHS-4; OR = 0.82 NFHS-5, p <0.01) as compared to the younger women (OR = 1). This may be due to childbearing which causes anemia and also some women attain menopause which improves Hb in blood. Theo dds of women having anemia were lower among all social group women (SC & ST, OBC, general) who were higher educated (OR = 0.93 SC & ST, OR = 0.93 OBC in NFHS-3; OR = 0.86 SC & ST, OR = 0.93 OBC, OR = 0.91 general in NFHS-4; OR = 0.87 SC & ST, OR = 0.90 OBC, OR = 0.89 general in NFHS-5) as compared to the no educated women. The chances of having anemia was less among SC & ST women in richest household (OR = 0.50 NFHS-3; OR = 0.72 NFHS-4; OR = 0.84 NFHS-5), richer household (OR = 0.59 NFHS-3; OR = 0.74 NFHS-4; OR = 0.90 NFHS-5), middle household (OR = 0.62 in NFHS-3; OR = 0.76 in NFHS-4; OR = 0.85 in NFHS-5), poor household (OR = 0.74 NFHS-3; OR = 0.83 NFHS-4; NFHS = 0.84 NFHS-5) as compared with the poorest household (OR = 1). Similar kinds of differences in the possibility of anemia prevalence were observed among the poorest to the richest group in OBC and general caste with a <0.01 level of significance. Thus economic condition remains one of the most important determinants of anemia among women.

The women in all social groups who were currently married were more likely to be anemic as compared to the never married women. Expectedly, the risk of anemia significantly

**Table 3. Prevalence of anemia among reproductive age-group women of different social group by background characteristics in India, 2015–16.**

| Background characteristics | NFHS 4 | | | | | | | | | | | |
| --- | --- | --- | --- | --- | --- | --- | --- | --- | --- | --- | --- | --- |
| | SC& ST | | | | OBC | | | | Others | | | |
| | Severe | Moderate | Mild | Any | Severe | Moderate | Mild | Any | Severe | Moderate | Mild | Any |
| *Age Group* | | | | | | | | | | | | |
| 15–24 | 1.21 | 14.71 | 42.23 | 58.14 | 0.99 | 12.14 | 39.53 | 52.66 | 0.71 | 10.83 | 39.13 | 50.67 |
| 25–34 | 1.08 | 14.09 | 41.39 | 56.56 | 0.98 | 12.31 | 38.48 | 51.76 | 0.66 | 10.50 | 37.88 | 49.04 |
| 35+ | 1.35 | 13.85 | 41.65 | 56.86 | 1.29 | 12.18 | 38.77 | 52.24 | 0.90 | 10.72 | 38.26 | 49.87 |
| χ2 | 99.51*** | | | 78.75*** | 46.24*** | | | 22.54*** | 32.11*** | | | 9.49*** |
| *Place of residence* | | | | | | | | | | | | |
| Urban | 1.16 | 13.57 | 39.90 | 54.62 | 1.01 | 11.79 | 38.02 | 50.82 | 0.72 | 9.97 | 37.74 | 48.43 |
| Rural | 1.24 | 14.45 | 42.39 | 58.08 | 1.13 | 12.42 | 39.42 | 52.98 | 0.80 | 11.29 | 38.99 | 51.08 |
| χ2 | 499.89*** | | | 499.2*** | 196.08*** | | | 191.43*** | 87*** | | | 61.79*** |
| *Educational status* | | | | | | | | | | | | |
| No education | 1.46 | 15.44 | 42.67 | 59.57 | 1.38 | 13.70 | 39.88 | 54.96 | 0.99 | 12.51 | 39.66 | 53.17 |
| Primary | 1.15 | 14.41 | 42.76 | 58.33 | 1.31 | 13.12 | 38.58 | 53.01 | 0.96 | 11.20 | 40.02 | 52.18 |
| Secondary & Higher | 1.07 | 13.32 | 40.88 | 55.27 | 0.91 | 11.29 | 38.57 | 50.77 | 0.68 | 10.20 | 37.90 | 48.78 |
| χ2 | 981.36*** | | | 931.45*** | 595.39*** | | | 396.78*** | 316.06*** | | | 179.47*** |
| *Wealth Index* | | | | | | | | | | | | |
| Poorest | 1.29 | 15.44 | 45.08 | 61.81 | 1.19 | 13.28 | 41.44 | 55.91 | 1.01 | 12.50 | 42.62 | 56.13 |
| Poorer | 1.29 | 14.38 | 42.01 | 57.68 | 1.24 | 12.89 | 39.95 | 54.08 | 0.79 | 11.31 | 41.06 | 53.16 |
| middle | 1.29 | 14.36 | 39.85 | 55.50 | 1.24 | 12.86 | 38.70 | 52.80 | 0.76 | 11.82 | 38.99 | 51.57 |
| Richer | 1.23 | 13.09 | 39.34 | 53.66 | 0.98 | 11.98 | 37.85 | 50.81 | 0.98 | 11.01 | 37.50 | 49.50 |
| Richest | 0.68 | 11.97 | 39.58 | 52.23 | 0.85 | 10.23 | 37.50 | 48.59 | 0.55 | 9.21 | 36.72 | 46.48 |
| χ2 | 2400*** | | | 2400*** | 694.9*** | | | 630.8*** | 353.2*** | | | 224.47*** |
| *Current work status* | | | | | | | | | | | | |
| Not working | 1.41 | 15.26 | 40.89 | 57.56 | 1.18 | 13.13 | 38.23 | 52.54 | 1.13 | 10.69 | 38.32 | 50.14 |
| Working | 1.61 | 15.83 | 41.74 | 59.18 | 1.54 | 14.71 | 38.03 | 54.28 | 1.05 | 11.51 | 35.56 | 48.13 |
| χ2 | 2.93 | | | 1.17 | 9.38** | | | 2.03 | 6.15 | | | 4.51** |
| *Marital status* | | | | | | | | | | | | |
| Not Married | 1.32 | 13.16 | 42.29 | 56.78 | 1.03 | 10.87 | 39.45 | 51.35 | 0.79 | 10.00 | 39.25 | 50.04 |
| Currently in union | 1.13 | 14.49 | 41.63 | 57.25 | 1.09 | 12.46 | 38.80 | 52.35 | 0.75 | 10.79 | 38.11 | 49.65 |
| FormalyMarried | 2.03 | 15.39 | 41.72 | 59.14 | 1.48 | 15.03 | 38.76 | 55.28 | 0.94 | 12.87 | 39.43 | 53.24 |
| χ2 | 163.07*** | | | 76.23*** | 212.96*** | | | 51.32*** | 75.5*** | | | 18.23*** |
| *Total Children ever born* | | | | | | | | | | | | |
| No Children | 1.36 | 13.73 | 41.26 | 56.35 | 1.04 | 11.43 | 38.19 | 50.67 | 0.75 | 10.05 | 37.99 | 48.78 |
| 1 to 2 children | 1.01 | 14.65 | 41.41 | 57.08 | 1.07 | 12.52 | 38.65 | 52.25 | 0.63 | 10.40 | 38.38 | 49.41 |
| > 2 Children | 1.30 | 14.25 | 42.62 | 58.17 | 1.17 | 12.55 | 40.04 | 53.77 | 1.01 | 11.95 | 39.03 | 51.99 |
| χ2 | 140.63*** | | | 95.24*** | 194.11*** | | | 152.15*** | 155.55*** | | | 83.1*** |
| *BMI* | | | | | | | | | | | | |
| Under weight | 1.85 | 16.30 | 44.41 | 62.57 | 1.56 | 14.14 | 41.36 | 57.06 | 1.22 | 13.33 | 40.83 | 55.38 |
| Normal | 1.12 | 14.04 | 41.62 | 56.77 | 1.03 | 12.16 | 39.08 | 52.27 | 0.77 | 10.80 | 38.86 | 50.43 |
| Over weight | 0.47 | 11.35 | 37.43 | 49.24 | 0.81 | 10.11 | 35.89 | 46.81 | 0.48 | 8.76 | 36.22 | 45.46 |
| Obese | 0.43 | 10.99 | 38.62 | 50.04 | 0.47 | 10.34 | 36.35 | 47.16 | 0.33 | 8.62 | 35.10 | 44.05 |
| χ2 | 2900*** | | | 2600*** | 1500*** | | | 1300*** | 786.48*** | | | 635.99*** |
| *Contraceptive use* | | | | | | | | | | | | |
| Not using | 1.35 | 14.94 | 41.09 | 57.37 | 1.09 | 12.42 | 38.43 | 51.93 | 0.79 | 10.81 | 37.88 | 49.48 |
| Pill & IUD | 0.70 | 11.84 | 43.99 | 56.53 | 0.87 | 10.81 | 40.87 | 52.55 | 0.61 | 8.66 | 39.67 | 48.94 |
| Female Sterilization | 1.10 | 13.26 | 42.31 | 56.68 | 1.15 | 12.22 | 39.27 | 52.65 | 0.81 | 10.80 | 38.47 | 50.08 |

*(Continued)*

**Table 3.** (Continued)

| Background characteristics | NFHS 4 | | | | | | | | | | | |
|---|---|---|---|---|---|---|---|---|---|---|---|---|
| | SC& ST | | | | OBC | | | | Others | | | |
| | Severe | Moderate | Mild | Any | Severe | Moderate | Mild | Any | Severe | Moderate | Mild | Any |
| Other | 0.94 | 13.72 | 44.02 | 58.69 | 1.00 | 10.99 | 41.03 | 53.03 | 0.65 | 10.85 | 40.25 | 51.75 |
| χ2 | 370.03*** | | | 192.56*** | 113.38*** | | | 17.41*** | 89.73*** | | | 61.63*** |
| *Mass media exposure* | | | | | | | | | | | | |
| No | 1.35 | 15.38 | 43.84 | 60.57 | 1.18 | 13.25 | 40.80 | 55.24 | 0.88 | 12.23 | 39.71 | 52.83 |
| Partial | 1.20 | 14.03 | 41.37 | 56.60 | 1.11 | 12.13 | 38.56 | 51.80 | 0.75 | 10.57 | 38.47 | 49.79 |
| everyday | 0.91 | 12.38 | 38.85 | 52.14 | 0.75 | 10.60 | 38.00 | 49.35 | 0.75 | 9.93 | 36.87 | 47.54 |
| χ2 | 1100*** | | | 1100*** | 504.19*** | | | 455.09*** | 225.54*** | | | 179.8*** |
| *Frequency of eating pulses* | | | | | | | | | | | | |
| Less than daily /not at al | 1.27 | 14.66 | 42.15 | 58.09 | 1.21 | 12.59 | 38.72 | 52.52 | 0.80 | 10.94 | 38.57 | 50.32 |
| Daily | 1.14 | 13.63 | 41.26 | 56.03 | 0.95 | 11.73 | 39.22 | 51.90 | 0.72 | 10.43 | 38.28 | 49.43 |
| | 103.0*** | | | 91.87*** | 65.30*** | | | 5.30** | 20.45*** | | | 0.26 |
| Source of Drinking water | | | | | | | | | | | | |
| Improved Source | 1.18 | 14.07 | 41.70 | 56.95 | 1.10 | 12.13 | 38.97 | 52.2 | 0.77 | 10.53 | 38.65 | 49.95 |
| Non-improved Souerce | 1.41 | 15.02 | 42.72 | 59.15 | 1.07 | 12.21 | 38.89 | 52.17 | 0.71 | 11.27 | 34.76 | 46.74 |
| χ2 | 12.80*** | | | 78.75*** | 19.09*** | | | 22.54*** | 126.95*** | | | 9.49*** |

*$p < .10$.

**$p < .05$.

***$p < .01$.

increases with an increase in the number of children ever born among reproductive women as repetative pregnancy reduces Hb in blood especially among the women who have lesser access and affordability to good food. The risk of anemia decrease among normal-weight women as normal weight is the expression of good health. Similar results were found in NFHS-4 & NFHS-5. Furthermore, the women who had more exposure to the mass media were less likely to be anemic as compared to those who either do not or have partial exposure.

## Discussion and conclusion

This study was carried out to understand the regional variation, trend of anemia prevalence and the influence of a wide range of socio-economic, demographic, and nutritional behaviour factors on the prevalence of anemia among different social groups of women of reproductive age in India.

Regional variation in the prevalence of anemia indicates that anemia is one of the major problems in the majority of the states of India. The changes from NFHS-3 to NFHS-5 show that the majority of the states reveal a very poor outcome in terms of anemia prevalence, and the prevalence rate increases over time. Apart from the high prevalence of anemia, the rate of prevalence varies over different geographical zones and states. SC & ST, OBC, and general women from eastern, north-eastern, and central zones suffer more from anemia than in other parts of India. The very high prevalence of anemia in all social groups has been noticed in Assam, Tripura from the north eastern zone; Bihar, Jharkhand, Odisha, West Bengal from eastern zone and Haryana from central zone from 2005–06 to 2019–21. The southern states are in much better condition than the other states of India. Except for Andhra Pradesh, other states namely, Tamil Nadu, Karnataka, and especially Kerala have much better conditions with

**Table 4. Prevalence of anemia level among women in different social group by background characteristics in India, 2019–21.**

| Background characteristics | NFHS 5 | | | | | | | | | | | |
|---|---|---|---|---|---|---|---|---|---|---|---|---|
| | SC& ST | | | | OBC | | | | Others | | | |
| | Severe | Moderate | Mild | Any | Severe | Moderate | Mild | Any | Severe | Moderate | Mild | Any |
| *Age Group* | | | | | | | | | | | | |
| 15–24 | 2.91 | 32.92 | 26.34 | 62.17 | 2.23 | 27.11 | 26.20 | 55.53 | 2.16 | 28.04 | 26.05 | 56.24 |
| 25–34 | 2.82 | 31.46 | 26.08 | 60.37 | 2.20 | 26.31 | 25.19 | 53.70 | 2.12 | 26.95 | 25.70 | 54.77 |
| 35+ | 3.60 | 31.53 | 25.07 | 60.20 | 2.98 | 26.92 | 24.46 | 54.37 | 2.86 | 26.91 | 25.40 | 55.17 |
| χ2 | 287.88*** | | | 195.6*** | 212.78*** | | | 58.95*** | 94.52*** | | | 54.6*** |
| *Place of residence* | | | | | | | | | | | | |
| Urban | 3.04 | 29.07 | 25.06 | 57.18 | 2.27 | 25.08 | 24.76 | 52.11 | 1.99 | 25.55 | 25.68 | 53.22 |
| Rural | 3.15 | 32.91 | 26.06 | 62.11 | 2.60 | 27.59 | 25.49 | 55.68 | 2.73 | 28.53 | 25.70 | 56.96 |
| χ2 | 620.02*** | | | 195.6*** | 441.86*** | | | 58.95*** | 208.77*** | | | 54.7*** |
| Educational status | | | | | | | | | | | | |
| No education | 3.26 | 33.07 | 25.90 | 62.23 | 2.98 | 28.77 | 24.89 | 56.64 | 3.28 | 28.91 | 25.92 | 58.12 |
| Primary | 3.68 | 33.47 | 25.67 | 62.81 | 2.83 | 28.14 | 25.20 | 56.17 | 2.91 | 28.69 | 26.23 | 57.83 |
| Secondary & Higher | 2.92 | 31.10 | 25.81 | 59.83 | 2.27 | 25.90 | 25.40 | 53.57 | 2.19 | 26.78 | 25.58 | 54.55 |
| χ2 | 506.30*** | | | 438.4*** | 337.07*** | | | 182.8*** | 186.99*** | | | 106.2*** |
| *Wealth Index* | | | | | | | | | | | | |
| Poorest | 2.85 | 35.82 | 27.45 | 66.12 | 2.73 | 30.69 | 26.40 | 59.82 | 3.17 | 32.88 | 27.06 | 63.11 |
| Poorer | 3.28 | 32.46 | 25.90 | 61.65 | 2.70 | 28.17 | 26.24 | 57.11 | 2.50 | 29.50 | 26.95 | 58.95 |
| Middle | 3.35 | 30.17 | 25.02 | 58.54 | 2.76 | 27.23 | 25.17 | 55.15 | 2.96 | 28.64 | 26.06 | 57.67 |
| Richer | 3.44 | 28.92 | 24.74 | 57.09 | 2.35 | 25.66 | 24.66 | 52.67 | 2.54 | 27.02 | 25.09 | 54.65 |
| Richest | 2.60 | 28.17 | 24.13 | 54.90 | 1.97 | 23.18 | 24.18 | 49.33 | 1.81 | 24.28 | 25.02 | 51.11 |
| χ2 | 1900*** | | | 1800*** | 1400*** | | | 1200*** | 535.32*** | | | 412.7*** |
| *Current work status* | | | | | | | | | | | | |
| Not working | 3.16 | 32.14 | 25.37 | 60.68 | 2.27 | 26.11 | 24.55 | 52.93 | 2.30 | 26.31 | 25.88 | 54.50 |
| Working | 3.89 | 32.17 | 24.20 | 60.27 | 2.94 | 26.87 | 24.55 | 54.35 | 2.67 | 27.77 | 23.86 | 54.31 |
| χ2 | 0.98 | | | 0.01 | 9.03 | | | 2.79* | 5.76 | | | 2.27 |
| *Marital status* | | | | | | | | | | | | |
| Not Married | 3.33 | 31.75 | 25.70 | 60.78 | 2.40 | 26.17 | 25.99 | 54.56 | 2.02 | 26.85 | 25.47 | 54.33 |
| Currently in union | 2.95 | 32.03 | 25.90 | 60.89 | 2.45 | 26.84 | 25.09 | 54.38 | 2.45 | 27.38 | 25.83 | 55.65 |
| Formaly Married | 4.60 | 32.61 | 25.16 | 62.37 | 3.86 | 29.86 | 24.13 | 57.84 | 4.38 | 27.61 | 24.48 | 56.46 |
| χ2 | 73.39*** | | | 1.80 | 121.89*** | | | 29.03*** | 35.13*** | | | 5.34* |
| *Total Children ever born* | | | | | | | | | | | | |
| No Children | 3.23 | 31.41 | 25.58 | 60.22 | 2.39 | 25.75 | 25.67 | 53.81 | 2.10 | 26.45 | 25.31 | 53.85 |
| 1 to 2 children | 2.90 | 32.38 | 26.11 | 61.39 | 2.41 | 26.74 | 25.07 | 54.21 | 2.35 | 27.37 | 26.04 | 55.76 |
| > 2 Children | 3.28 | 32.11 | 25.71 | 61.09 | 2.74 | 28.01 | 25.11 | 55.86 | 2.96 | 28.13 | 25.48 | 56.58 |
| χ2 | 34.38*** | | | 13.57*** | 110.78*** | | | 44.35*** | 65.08*** | | | 15.93*** |
| *BMI* | | | | | | | | | | | | |
| Under weight | 4.44 | 36.76 | 25.07 | 66.27 | 3.76 | 31.00 | 25.24 | 60.01 | 3.80 | 32.01 | 25.08 | 60.88 |
| Normal | 2.92 | 31.73 | 26.50 | 61.15 | 2.42 | 26.73 | 25.71 | 54.86 | 2.30 | 27.16 | 26.02 | 55.48 |
| Over weight | 2.18 | 27.29 | 24.65 | 54.11 | 1.78 | 23.52 | 24.33 | 49.62 | 1.90 | 25.10 | 25.63 | 52.62 |
| Obese | 2.36 | 27.32 | 24.27 | 53.96 | 1.48 | 24.39 | 24.02 | 49.89 | 2.13 | 25.13 | 24.87 | 52.13 |
| χ2 | 3000*** | | | 2400*** | 1800*** | | | 1300*** | 564.57*** | | | 353.6*** |
| *Contraceptive use* | | | | | | | | | | | | |
| Not using | 3.15 | 31.96 | 25.62 | 60.73 | 2.35 | 26.33 | 25.63 | 54.31 | 2.29 | 26.88 | 25.40 | 54.57 |
| Pill & IUD | 2.26 | 32.77 | 27.52 | 62.55 | 1.99 | 27.62 | 26.39 | 56.00 | 2.38 | 29.05 | 26.92 | 58.35 |
| Female Sterilization | 3.33 | 31.37 | 25.45 | 60.15 | 2.88 | 27.08 | 24.32 | 54.28 | 2.88 | 27.54 | 25.51 | 55.93 |

(*Continued*)

**Table 4.** (Continued)

| Background characteristics | NFHS 5 | | | | | | | | | | | |
| --- | --- | --- | --- | --- | --- | --- | --- | --- | --- | --- | --- | --- |
| | SC& ST | | | | OBC | | | | Others | | | |
| | Severe | Moderate | Mild | Any | Severe | Moderate | Mild | Any | Severe | Moderate | Mild | Any |
| Other | 2.88 | 33.20 | 26.71 | 62.79 | 2.32 | 27.56 | 25.71 | 55.58 | 2.12 | 27.29 | 26.26 | 55.68 |
| χ2 | 148.82*** | | | 103.4*** | 131.00*** | | | 36.19*** | 47.91*** | | | 24.52*** |
| *Mass media exposure* | | | | | | | | | | | | |
| No | 3.04 | 33.99 | 26.61 | 63.64 | 2.54 | 28.85 | 25.75 | 57.14 | 2.82 | 29.73 | 26.49 | 59.04 |
| Partial | 3.17 | 31.33 | 25.61 | 60.10 | 2.54 | 26.47 | 25.17 | 54.17 | 2.36 | 27.15 | 25.67 | 55.18 |
| everyday | 3.02 | 30.10 | 24.43 | 57.56 | 2.01 | 24.25 | 24.78 | 51.04 | 2.25 | 24.39 | 24.65 | 51.29 |
| χ2 | 609.27*** | | | 551.7*** | 320.65*** | | | 274.8*** | 223.60*** | | | 183.1*** |
| *Frequency of eating pulses* | | | | | | | | | | | | |
| Less than daily /not at al | 3.33 | 32.55 | 25.38 | 61.26 | 2.76 | 27.29 | 24.96 | 55.00 | 2.70 | 27.84 | 25.23 | 55.78 |
| Daily | 2.88 | 31.35 | 26.32 | 60.56 | 2.24 | 26.33 | 25.56 | 54.13 | 2.15 | 26.71 | 26.13 | 54.99 |
| χ2 | 87.89*** | | | 54.36*** | 84.82*** | | | 8.69*** | 154.50*** | | | 51.40*** |
| Source of Drinking water (Improved source®) | | | | | | | | | | | | |
| Improved Source | 3.17 | 31.82 | 25.79 | 60.78 | 2.51 | 26.69 | 25.24 | 54.44 | 2.43 | 27.30 | 25.70 | 55.43 |
| Non-improved Souerce | 2.72 | 33.10 | 26.48 | 62.30 | 2.42 | 28.34 | 25.44 | 56.20 | 2.29 | 27.13 | 24.92 | 54.34 |
| χ2 | 32.72*** | | | 12.41*** | 8.42** | | | 2.86* | 31.87*** | | | 31.70** |

respect to anemia prevalence of women in all social groups. Similar results were observed in the previous studies in India [40, 41].

Results from NFHS-3, NFHS-4, and NFHS-5 suggest that the prevalence of anemia rate increased in recent times and it is not uniformly distributed, rather it is skewed towards socially disadvantaged groups of women from SC & ST, OBC who are more prone to anemia than general women across the variables. The socially disadvantaged women who are living in rural areas were more prone to anemia. Whereas, the differences in anemia prevalence among SC & ST and general or OBC and general is high in urban areas than the rural areas. Studies observed that in India, urbanisation poses formidable challenges for controlling economic disparities and rising health inequalities [16, 22, 23]. Rural women suffer more probably because they have less access to health information, poor education, and less purchasing power to fulfill their basic requirement of quality food. The result showed a negative relation between the prevalence of anemia and the education level of women in all social groups, although socially and economically backward people were more at risk of anemia. This finding was consistent with the other studies conducted in China, and the Vellore district in India, where a higher prevalence of anemia was reported in lower-educated women [25, 33]. Bivariate and multivariate analysis showed the socioeconomic status of households was significantly associated with anemia in all social groups of women. In all social categories (ST & SC, OBC, and general) women from lower socio-economic categories have a higher prevalence of anemia than those from higher economic status. In India, the socially disadvantaged group is considered the most deprived, and that draws the attention of the government and researchers, but this result shows that the poorest and poor group of general caste women had worst conditions in anemia prevalence than the women who enjoyed better wealth quintile in SC & ST and OBC because the economy is one of the important determining factors for health, education, and livelihood. Like previous studies in other countries [27, 28] findings from this study support that poor women are associated with a high risk of anemia, irrespective of caste.

This study found an association of anemia with women's age. Prevalence of anemia was found to decrease with increases in age. Previous studies performed in developing countries of

**Table 5. Logistic regression estimates: Factors associated with anemia among different scial group of reproductive women in India, 2005–2006 & 2015–2016.**

| BackgroundCharacteristics | NFHS 3 | | | | | | NFHS 4 | | | | | |
|---|---|---|---|---|---|---|---|---|---|---|---|---|
| | SC & ST | | OBC | | General | | SC & ST | | OBC | | General | |
| | RR | 95% CI | RR | 95% CI | RR | 95% CI | RR | 95% CI | RR | 95% CI | RR | 95% CI |
| Age Group (15–24®) | | | | | | | | | | | | |
| 25–34 | .85*** | .796–.914 | .89*** | .833–.949 | .83*** | .78–.885 | .9*** | .846–.957 | .82*** | .774–.875 | .94 | .863–1.02 |
| 35+ | .87*** | .807–.946 | .90*** | .84–.973 | .84*** | .784–.905 | .86*** | .797–.919 | .79*** | .741–.85 | .88*** | .803–.968 |
| Place of residence (Urban®) | | | | | | | | | | | | |
| Rural | .93** | .881–985 | .98 | .928–1.01 | .87*** | .827–.908 | 1.02 | .963–1.07 | 1.0 | .952–1.04 | 1.0 | .943–1.06 |
| Educational status (No education ®) | | | | | | | | | | | | |
| Primary | .99 | .92–1.06 | 1.04 | .971–1.11 | 1.0 | .934–1.08 | .86*** | .81–.923 | .94** | .878–.999 | .94 | .847–1.04 |
| Secondary & Higher | .93** | .864–.992 | .92** | .866–.984 | .98 | .916–1.04 | .86*** | .816–.915 | .93*** | .878–.979 | .91*** | .838–.991 |
| Wealth Index (Poorest ®) | | | | | | | | | | | | |
| Poorer | .74*** | .684–.794 | .97 | .894–1.05 | .92 | .823–1.04 | .83*** | .785–.884 | .95 | .892–1.01 | .83*** | .737–.929 |
| middle | .62*** | .578–.675 | .93** | .856–1.01 | .78*** | 0.69–0.87 | .76*** | .716–.817 | .89*** | .834–.955 | .81*** | .718–.907 |
| Richer | .59*** | .544–.647 | .87*** | .798–.947 | .71*** | 0.63–0.79 | .74*** | .686–.798 | .92*** | .86–.994 | .79*** | .701–0.89 |
| Richest | .50*** | .457–.557 | .78*** | .706–.854 | .62*** | 0.55–0.69 | .72*** | .658–.787 | .81*** | .745–.873 | .77*** | .68–.869 |
| Current work status (Not working ®) | | | | | | | | | | | | |
| Working | .87*** | .829–.912 | .97 | .929–1.02 | .96* | 0.91–1.00 | 1.0 | .959–1.05 | 1.02* | .97–1.06 | .93** | .874–.999 |
| Marital status (Never Married ®) | | | | | | | | | | | | |
| Currently in union | 1.16*** | 1.05–1.27 | 1.12** | 1.03–1.22 | 1.11** | 1.02–1.21 | 1.03 | .947–1.12 | .97 | .895–1.05 | .92 | .827–1.03 |
| Formaly Married | 1.22*** | 1.06–1.39 | 1.14** | 1.00–1.31 | 1.17** | 1.03–1.34 | 1.02 | .906–1.16 | 1.15** | 1.02–1.30 | .92 | .778–1.09 |
| Total Children ever born (No Children ®) | | | | | | | | | | | | |
| 1 to 2 chil | 1.17*** | 1.06–1.29 | 1.2*** | 1.10–1.32 | 1.28*** | 1.17–1.39 | 1.13*** | 1.04–1.22 | 1.27*** | 1.17–1.37 | 1.2*** | 1.07–1.34 |
| > 2 Children | 1.12** | 1.01–1.24 | 1.3*** | 1.16–1.40 | 1.42*** | 1.29–1.56 | 1.1*** | 1.03–1.23 | 1.3*** | 1.22–1.45 | 1.3*** | 1.16–1.49 |
| BMI (Underweight®) | | | | | | | | | | | | |
| Normal | .77*** | 0.73–0.81 | .86*** | .822–.904 | .81*** | .767–.847 | .74*** | .705–.777 | .82*** | 0.78–0.85 | .81*** | .751–.866 |
| Over weight | .58*** | 0.53–0.63 | .66*** | .608–.712 | .61*** | .571–.657 | .49*** | .456–.531 | .69*** | 0.65–.739 | .7*** | .639–.765 |
| Obese | .58*** | .49–.694 | .61*** | .536–.691 | .67*** | .607–.741 | .63*** | .551–.711 | .69*** | .623–.757 | .63*** | .559–.709 |
| Contraceptive use (Not using®) | | | | | | | | | | | | |
| Pill & IUD | .63*** | .555–.709 | .76*** | .672–.862 | .78*** | .714–.862 | .89** | .804–.975 | 1.03 | .922–1.15 | .95 | .845–1.07 |
| Female Sterilization | .85*** | .794–.901 | .84*** | .792–.891 | .92*** | .863–.973 | 1.03 | .974–1.09 | 1.01 | .962–1.06 | 1.02 | .948–1.10 |
| Other | 1 | .919–1.10 | .93* | .856–1.01 | 1.03 | .961–1.20 | 1.2*** | 1.11 1.30 | 1.03 | .958–1.10 | 1.17*** | 1.08–1.27 |
| Mass media exposure (No ®) | | | | | | | | | | | | |
| Partial | .94* | .882–1.01 | .96 | .898–1.02 | .94* | .867–1.01 | 1.02 | .967–1.09 | .98 | .929–1.04 | 1.05 | .954–1.15 |
| everyday | .84*** | .766–.915 | .92* | .849–1.00 | .85*** | .776–.925 | .96 | .874 1.05 | .92** | .844–1.01 | .98 | .87–1.10 |
| Frequency of eating pulses (Less than daily/not at al®) | | | | | | | | | | | | |
| Daily | 1.16*** | 1.09–1.21 | 1.08 | 1.03–1.13 | 1.08** | 1.04–1.14 | 1.1*** | 1.05–1.14 | .97*** | .937–1.01 | 1.0*** | .967–1.07 |
| Source of drinking water (Improved source®) | | | | | | | | | | | | |
| Unmproved Source | 1.09*** | 1.03–1.17 | 0.95 | 0.893–1.02 | .85*** | 0.794–910 | 1.04** | 0.98–1.11 | .86 | .807–.926 | .92** | .836–1.01 |
| Cons | 2.7*** | 2.47–3.04 | 1.5*** | 1.31–1.61 | 1.8*** | 1.55–1.98 | 2.0*** | 1.94–2.33 | 1.5*** | 1.41–1.67 | 1.5*** | 1.30–1.71 |
| LR chi2(24) | | 1156.05 | | 537.40 | | 835.54 | | 879.56 | | 43363 | | 216.24 |
| Prob > chi2 | | 0 | | 0 | | 0 | | 0 | | 0 | | 0 |
| Pseudo R2 | | 0.0273 | | 0.0113 | | 0.0158 | | 0.0161 | | 0.007 | | 0.0066 |
| Number of obs. | | 30870 | | 34368 | | 38109 | | 39798 | | 44131 | | 23802 |

CI—confidence interval

®: Reference category;

P value:

*p < .10.

**p < .05.

***p < .01.

**Table 6. Logistic regression estimates: Factors associated with anemia among different scial group of reproductive women in India, 2019–2021.**

| Background Characteristics | SC | | OBC | | General | |
|---|---|---|---|---|---|---|
| | RR | 95% CI | RR | 95% CI | RR | 95% CI |
| Age Group (15–24®) | | | | | | |
| 25–34 | 0.84*** | 0.79–0.9 | 0.87*** | 0.81–0.93 | 0.84*** | 0.76–0.92 |
| 35+ | 0.75*** | 0.691–0.804 | 0.87*** | 0.8–0.94 | 0.82*** | 0.74–0.92 |
| Place of residence (Urban®) | | | | | | |
| Rural | 1.11*** | 1.05–1.18 | 1.08 | 1.03–1.14 | 1.11*** | 1.04–1.19 |
| Educational status (No education ®) | | | | | | |
| Primary | 1 | 0.93–1.08 | 0.99 | 0.92–1.06 | 0.94 | 0.83–1.06 |
| Secondary & Higher | 0.87*** | 0.82–0.92 | 0.90*** | 0.85–0.96 | 0.89** | 0.81–0.98 |
| Wealth Index (Poorest ®) | | | | | | |
| Poorer | 0.84*** | 0.79–0.89 | 0.97 | 0.91–1.04 | 0.87** | 0.77–0.99 |
| middle | 0.85*** | 0.8–0.91 | 0.90*** | 0.84–0.96 | 0.84*** | 0.74–0.95 |
| Richer | 0.9*** | 0.84–0.97 | 0.89*** | 0.83–0.96 | 0.92 | 0.81–1.04 |
| Richest | 0.84*** | 0.77–0.92 | 0.82*** | 0.75–0.89 | 0.82*** | 0.72–0.94 |
| Current work status (Not working ®) | | | | | | |
| Working | 1.01 | 0.97–1.06 | 1.03 | 0.98–1.08 | 0.95 | 0.89–1.03 |
| Marital status (Never Married ®) | | | | | | |
| Currently in union | 1.04 | 0.95–1.14 | 0.92** | 0.84–1.0 | 0.98 | 0.86–1.12 |
| Formaly Married | 1.11 | 0.98–1.26 | 1.03 | 0.9–1.17 | 1.05 | 0.86–1.28 |
| Total Children ever born (No Children ®) | | | | | | |
| 1 to 2 chil | 1.14*** | 1.04–1.24 | 1.26 | 1.15–1.37 | 1.15** | 1.01–1.31 |
| > 2 Children | 1.11* | 1.01–1.23 | 1.26 | 1.14–1.39 | 1.14** | 0.99–1.32 |
| BMI (Underweight®) | | | | | | |
| Normal | 0.69*** | 0.65–0.73 | 0.82*** | 0.77–0.86 | 0.85*** | 0.78–0.93 |
| Over weight | 0.55*** | 0.51–0.59 | 0.67*** | 0.63–0.72 | 0.71*** | 0.64–0.79 |
| Obese | 0.59*** | 0.52–0.66 | 0.72*** | 0.66–0.8 | 0.83*** | 0.72–0.94 |
| Contraceptive use (Not using®) | | | | | | |
| Pill & IUD | 0.94 | 0.86–1.03 | 1.03 | 0.93–1.14 | 1.04 | 0.92–1.19 |
| Female Sterilization | 1.08*** | 1.02–1.15 | 0.98 | 0.92–1.04 | 1.03 | 0.94–1.12 |
| Other | 1.11*** | 1.04–1.19 | 1.07 | 1–1.14 | 1.11 | 1.02–1.21 |
| Mass media exposure (No ®) | | | | | | |
| Partial | 1.04 | 0.99–1.1 | 1.04 | 0.98–1.1 | 0.94 | 0.86–1.03 |
| everyday | 0.96 | 0.87–1.06 | 0.99 | 0.9–1.09 | 0.98 | 0.86–1.11 |
| Frequency of eating pulses (Less than daily/not at al®) | | | | | | |
| Daily | 1.05** | 1.01–1.1 | 0.99 | 0.96–1.03 | 0.93** | 0.88–0.98 |
| Source of drinking water ((Improved source®)) | | | | | | |
| Unmproved Source | 0.97 | 0.90–1.04 | 0.94 | 0.86–1.02 | 0.85** | 0.73–0.95 |
| Cons | 2.05*** | 1.86–2.26 | 1.51*** | 1.36–1.66 | 1.82 | 1.54–2.14 |
| LR chi2(24) | | 612.90 | | 354.89 | | 163.08 |
| Prob > chi2 | | 0 | | 0 | | 0 |
| Pseudo R2 | | 0.012 | | 0.0067 | | 0.0063 |
| Number of obs. | | 37342 | | 38550 | | 18682 |

CI—confidence interval

®: Reference category;

P value:

*p < .10.

**p < .05.

***p < .01.

Africa and Asia document the association between women age and anemia [22, 24, 41]. The majority of women at a younger age may be anemic due to lower dietary iron intake and the additional demand for iron imposed by loss of iron during menstruation, pregnancy, and lactation [16]. This study observed a similar kind of association between social group, anemia and BMI. SC & ST women who are majorly poor are more likely to be anemic and to be underweight than general women. Previous research studies observed similar kind of scenario, where researchers describe that the women in SC & ST group were mostly living below poverty line and more likely to be thin or underweight, and general women enjoy a better quality of life and often are likely to be overweight and obese [11, 42]. However, this study reveals the effect of poverty on women's nutrition and anemia prevalence is uniform regardless of social categories.

After examining the overall findings of this study, it is observed that different sociodemographic factors are responsible for predicting anemia levels among the social groups of women in India. The factors responsible for anemia among different social groups are multiple, ranging from poor economic and educational status, rural residence to the exposure of mass media of women. This study indicates that economic conditions (wealth quintile) dominantly control the anemia level in all social groups. Economic condition determines the quantity, quality and variety of food which in turn take care of over all nutrition level as well as anemia. Though the proportion remains low, but variation across social group is significant while considering the frequency of pulse intake.

This is evident from the above discussion that the problem of iron deficiency anemia remain a major problem in India. Research from previous studies suggests that for eradicating anemia, India should improve women's overall nutrition status and their income so that they have better access to the resources [43]. Government of India launched different programmes to eradicate the anemia problem at different times. Ministry of Health and Family Welfare first launched National Nutritional Anaemia Prophylaxis Programme (NNAPP) in 1970 to reduce the iron deficiency of women and children [44, 45]. Further, India government launched the Anemia Mukt Bharat program in 2018 with an ambitious target of reducing anemia prevalence among reproductive age group women to 35%. The above analysis helped to understand the essential inputs on strengthening the programme to make the programme successful [46]. Emphasising on less number of births, improving BMI through diet including pulse intake probably would be able to address the problem to some extent. Nutrition is a study that requires to address the health condition available to the population, including safe drinking water, sanitation, dietary diversity etc. NFHS has ceratin limitation to provide data on these aspects and thus leaves scope for further studies across the social groups.

## Acknowledgments

We thank two anonymous reviewers for the excellent and positive suggestions that helped to improve this paper.

## Author Contributions

**Conceptualization:** Nowaj Sharif, Bhaswati Das, Asraful Alam.

**Data curation:** Nowaj Sharif.

**Formal analysis:** Nowaj Sharif, Asraful Alam.

**Investigation:** Nowaj Sharif, Bhaswati Das.

**Methodology:** Nowaj Sharif, Asraful Alam.

**Project administration:** Bhaswati Das.

**Software:** Nowaj Sharif.

**Supervision:** Bhaswati Das, Asraful Alam.

**Validation:** Nowaj Sharif.

**Visualization:** Nowaj Sharif, Bhaswati Das, Asraful Alam.

**Writing – original draft:** Nowaj Sharif, Asraful Alam.

**Writing – review & editing:** Bhaswati Das, Asraful Alam.

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
