## [Decision Letter · Decision Letter 0]

17 Nov 2022

PONE-D-22-28727Prevalence of Anemia among reproductive women in different Social Group in India:PLOS ONE

Dear Dr. Das,

Thank you for submitting your manuscript to PLOS ONE. After careful consideration, we feel that it has merit but does not fully meet PLOS ONE’s publication criteria as it currently stands. Therefore, we invite you to submit a revised version of the manuscript that addresses the points raised during the review process. Kindly go through the review comments by both the reviewers and revise your paper carefully.  Please submit your revised manuscript by Jan 01 2023 11:59PM. If you will need more time than this to complete your revisions, please reply to this message or contact the journal office at plosone@plos.org. Please include the following items when submitting your revised manuscript:A rebuttal letter that responds to each point raised by the academic editor and reviewer(s). You should upload this letter as a separate file labeled 'Response to Reviewers'.A marked-up copy of your manuscript that highlights changes made to the original version. You should upload this as a separate file labeled 'Revised Manuscript with Track Changes'.An unmarked version of your revised paper without tracked changes. You should upload this as a separate file labeled 'Manuscript'.

We look forward to receiving your revised manuscript.

Kind regards,

Vijayaprasad Gopichandran

Academic Editor

PLOS ONE

Journal Requirements:

5. We note that Figure 1 in your submission contain map images which may be copyrighted. All PLOS content is published under the Creative Commons Attribution License (CC BY 4.0), which means that the manuscript, images, and Supporting Information files will be freely available online, and any third party is permitted to access, download, copy, distribute, and use these materials in any way, even commercially, with proper attribution. For these reasons, we cannot publish previously copyrighted maps or satellite images created using proprietary data, such as Google software (Google Maps, Street View, and Earth). For more information, see our copyright guidelines: http://journals.plos.org/plosone/s/licenses-and-copyright.

(1) You may seek permission from the original copyright holder of Figure 1 to publish the content specifically under the CC BY 4.0 license.  

Reviewers' comments:

Reviewer's Responses to Questions

**Comments to the Author**

1. Is the manuscript technically sound, and do the data support the conclusions?

Reviewer #1: Yes

Reviewer #2: No

2. Has the statistical analysis been performed appropriately and rigorously? 

Reviewer #1: Yes

Reviewer #2: I Don't Know

3. Have the authors made all data underlying the findings in their manuscript fully available?

Reviewer #1: Yes

Reviewer #2: Yes

4. Is the manuscript presented in an intelligible fashion and written in standard English?

Reviewer #1: Yes

Reviewer #2: No

5. Review Comments to the Author

Reviewer #1: Authors many incorporate following suggestions to improve their paper

1. Authors may include variables related to water sanitation and hygiene in their logistic regression model. Studies have shown that prevalence of anemia is higher among women who have who used unsafe water compared to women who have used safe water and sanitation facility

2. Authors may consider the pooled data set in the logistics regression model to know whether certain determinants effects changed/remained the same over 10-19 years period.

3. Although authors have mentioned about sample size issue but pregnancy status is important to understand the anemia status of the women. Pooled data may solve this thin sample issue.

Reviewer #2: Dear Author,

Thank you for choosing me to review the following manuscript.

Prevalence of anaemia is wide spread in India and most of Low and Middle Income counties. (LMICs). With respect to India, the large scale National survey (NFHS 1,2,3,4,5) is major source to determine health status of Indian population including Anaemia. Based on these NFHS and CNNS survey data most of the National Public polices are framed and implemented. one such recent initiative is Anaemia mukt Bharath from ministry of Health and Family welfare, GOI, to reduce its prevalence particularly in WRA. There are many reports and articles based on these surveys, which clearly mentions the different social-economic and associated factors responsible for its prevalence among WRA and also possible measures to overcome the burden of anaemia in Indian subcontinent (please see the below articles in the similar line).

In this context, the manuscript titled "Prevalence of Anemia among reproductive women in different Social Group in India" is not adding any new information to the existing literature. Though the manuscript included the recent NFHS-5 survey data for all the analysis, no new information/outcome was emerged from this new analysis.

literature for reference

doi: 10.1177/1010539512442567

doi:10.4103/2224-3151.228423

doi.org/10.1017/S0021932009990149

doi.org/10.1038/s41591-021-01498-0

And the recent review paper https://doi.org/10.3390/nu13082745 that aimed to compile evidence on the determinants and drivers of WRA anaemia reduction in low- and middle-income countries (LMICs).

And also Please see the following perspective the way forward to overcome the anaemia burden in India.

Kurpad AV, Sachdev HS. Childhood and Adolescent Anemia Burden in India: The Way Forward. Indian Pediatrics. 2022 Aug 26:S097475591600447-.

Thank you

Sincerely,

Rajashekar Reddy

6. PLOS authors have the option to publish the peer review history of their article (what does this mean?). If published, this will include your full peer review and any attached files.

Reviewer #1: No

Reviewer #2: No

---

## [Author Response · Author response to Decision Letter 0]

11 Jan 2023

PONE-D-22-28727

Prevalence of Anemia among reproductive women in different Social Group in India:

PLOS ONE

Dear Dr. Das,

Thank you for submitting your manuscript to PLOS ONE. After careful consideration, we feel that it has merit but does not fully meet PLOS ONE’s publication criteria as it currently stands. Therefore, we invite you to submit a revised version of the manuscript that addresses the points raised during the review process.

Kindly go through the review comments by both the reviewers and revise your paper carefully. 

We look forward to receiving your revised manuscript.

Kind regards,

Vijayaprasad Gopichandran

Academic Editor

PLOS ONE

Journal Requirements:

Answer: Modified According to the PLOS ONE MANUSCRIPT BODY FORMATTING GUIDELINES 

Answer: Added in Data Availability statement

Answer: done

Answer: done

5. We note that Figure 1 in your submission contain map images which may be copyrighted. All PLOS content is published under the Creative Commons Attribution License (CC BY 4.0), which means that the manuscript, images, and Supporting Information files will be freely available online, and any third party is permitted to access, download, copy, distribute, and use these materials in any way, even commercially, with proper attribution. For these reasons, we cannot publish previously copyrighted maps or satellite images created using proprietary data, such as Google software (Google Maps, Street View, and Earth). For more information, see our copyright guidelines: http://journals.plos.org/plosone/s/licenses-and-copyright.

(1) You may seek permission from the original copyright holder of Figure 1 to publish the content specifically under the CC BY 4.0 license. 

Answer: Modified and New Map added 

Reviewers' comments:

Reviewer's Responses to Questions

Comments to the Author

1. Is the manuscript technically sound, and do the data support the conclusions?

Reviewer #1: Yes

Reviewer #2: No

2. Has the statistical analysis been performed appropriately and rigorously?

Reviewer #1: Yes

Reviewer #2: I Don't Know

3. Have the authors made all data underlying the findings in their manuscript fully available?

Reviewer #1: Yes

Reviewer #2: Yes

4. Is the manuscript presented in an intelligible fashion and written in standard English?

Reviewer #1: Yes

Reviewer #2: No

5. Review Comments to the Author

Reviewer #1: Authors many incorporate following suggestions to improve their paper

1. Authors may include variables related to water sanitation and hygiene in their logistic regression model. Studies have shown that prevalence of anemia is higher among women who have who used unsafe water compared to women who have used safe water and sanitation facility

Answer: Modified accordingly 

2. Authors may consider the pooled data set in the logistics regression model to know whether certain determinants effects changed/remained the same over 10-19 years period.

3. Although authors have mentioned about sample size issue but pregnancy status is important to understand the anemia status of the women. Pooled data may solve this thin sample issue.

Answer: Modified accordingly and pooled data added

Reviewer #2: Dear Author,

Thank you for choosing me to review the following manuscript.

Prevalence of anaemia is wide spread in India and most of Low and Middle Income counties. (LMICs). With respect to India, the large scale National survey (NFHS 1,2,3,4,5) is major source to determine health status of Indian population including Anaemia. Based on these NFHS and CNNS survey data most of the National Public polices are framed and implemented. one such recent initiative is Anaemia mukt Bharath from ministry of Health and Family welfare, GOI, to reduce its prevalence particularly in WRA. There are many reports and articles based on these surveys, which clearly mentions the different social-economic and associated factors responsible for its prevalence among WRA and also possible measures to overcome the burden of anaemia in Indian subcontinent (please see the below articles in the similar line).

In this context, the manuscript titled "Prevalence of Anemia among reproductive women in different Social Group in India" is not adding any new information to the existing literature. Though the manuscript included the recent NFHS-5 survey data for all the analysis, no new information/outcome was emerged from this new analysis.

literature for reference

doi: 10.1177/1010539512442567

doi:10.4103/2224-3151.228423

doi.org/10.1017/S0021932009990149

doi.org/10.1038/s41591-021-01498-0

And the recent review paper https://doi.org/10.3390/nu13082745 that aimed to compile evidence on the determinants and drivers of WRA anaemia reduction in low- and middle-income countries (LMICs).

And also Please see the following perspective the way forward to overcome the anaemia burden in India.

Kurpad AV, Sachdev HS. Childhood and Adolescent Anemia Burden in India: The Way Forward. Indian Pediatrics. 2022 Aug 26:S097475591600447-.

Answer: Followed this literature carefully and modified accordingly

Thank you

Sincerely,

Rajashekar Reddy

6. PLOS authors have the option to publish the peer review history of their article (what does this mean?). If published, this will include your full peer review and any attached files.

Do you want your identity to be public for this peer review? For information about this choice, including consent withdrawal, please see our Privacy Policy.

Reviewer #1: No

Reviewer #2: No

---

## [Editor Report · Decision Letter 1]

13 Jan 2023

Prevalence of Anemia among reproductive women in different Social Group in India: Cross-sectional study using nationally representative data

PONE-D-22-28727R1

Dear Dr. Das,

We’re pleased to inform you that your manuscript has been judged scientifically suitable for publication and will be formally accepted for publication once it meets all outstanding technical requirements.

Kind regards,

Vijayaprasad Gopichandran

Academic Editor

PLOS ONE
---

## [Editor Report · Acceptance letter]

23 Jan 2023

PONE-D-22-28727R1 

Prevalence of Anemia among reproductive women in different Social Group in India: Cross-sectional study using nationally representative data 

Dear Dr. Das:

I'm pleased to inform you that your manuscript has been deemed suitable for publication in PLOS ONE. Congratulations! Your manuscript is now with our production department. 

Kind regards, 

on behalf of

Dr. Vijayaprasad Gopichandran 

Academic Editor

PLOS ONE